# An Analytical Model to Predict Foot Sole Temperature: Implications to Insole Design for Physical Activity in Sport and Exercise

Hossain Nemati [1,*] and Roozbeh Naemi [2]

1    Department of Mechanics, Marvdasht Branch, Islamic Azad University, Marvdasht 73711-13119, Iran
2    Centre for Biomechanics and Rehabilitation Technologies, Staffordshire University, Leek Road, Stoke-on-Trent ST4 2DF, UK; r.naemi@staffs.ac.uk
*    Correspondence: h.nemati@miau.ac.ir

**Abstract:** Foot sole temperature, besides its importance in thermal comfort, can be considered an important factor in identifying tissue injuries due to heavy activities or diseases. Hyperthermia, which is a raise in the foot temperature, increases the risk of diabetic ulcers considerably. In this study, a model is proposed to predict the foot sole temperature with acceptable accuracy. This model for the first time considers both the thermal and mechanical properties of the shoe sole, the intensity of the activity, the ambient condition, and sweating, which are involved in the thermal interaction between the sole of the foot and footwear. Furthermore, the proposed model provides the opportunity to estimate the contributions of different parameters in foot thermal regulation by describing the interaction of activity, duration, and intensity as well as sweating in influencing the foot sole temperature. In doing so it takes into account the relative importance of heat capacitance and the thermal conductivity. The results of this study revealed that sweating is not as effective in cooling the ball area of the foot while it is the principal contributor to thermal regulation in the arch area. The model also showed the importance of trapped air in keeping the foot warm, especially in cold conditions. Based on the simulation results, in selecting the shoe sole, and in addition to the conductivity, the thermal capacity of the sole of the shoe needs to be considered. The developed analytical model allowed the investigation of the contribution of all the involved parameters in foot thermal regulation and has shown that a different foot temperature can be achieved when the amount of material versus air is changed in the insole design. This can have practical implications in the insole design for a variety of conditions such as hypo and hyper-thermia in physical activities in sports and exercise settings.

**Keywords:** footwear; thermal comfort; foot sole temperature; sweating, plantar soft tissue

## 1. Introduction

In the design of footwear, usually, aesthetics and comfort are the main important factors considered, and thermal characteristics are not commonly taken into account. Hence, limited studies can be found that have a focus on the thermal characteristics of the footwear and the foot thermal regulation in a shoe [1–5]. The majority of the studies about footwear heat transfer deal with keeping feet warm in cold conditions [1,4,6–9]. Generally, feet are kept warm using thermal insulation or an electric heater pad to inhibit or compensate for the heat loss from the foot sole to the environment. However, the other aspect of foot thermal regulation involves cooling. The human body's mechanical efficiency is not more than 20~25% [10] and the rest of the energy dissipates to the environment in the form of heat. Consequently, the foot as a body thermal radiator [11] is responsible for dissipating heat. It was observed that in vigorous physical activities such as running, the temperature of the midsole can reach 50 °C or higher during a summer daytime [12].

Cooling feet is important for two reasons: first, thermal comfort is related mainly to skin temperature [13], and for feet, this comfort depends more on skin temperature rather than moisture [14]. The second and even more important reason, is that excess foot sole temperature is associated with diabetic foot ulcers [15–17].

Few experimental methods to measure the foot temperature and footwear thermal insulation assessment exist [2,3,5,18,19]. To assess the footwear thermal insulation, a plastic bag filled with hot water is usually placed inside the shoe to occupy the shoe space perfectly, and then, the temperatures of the shoe's different parts are measured during the time [5,20]. Some other experiments involve measuring the temperature in situ, by thermocouples during walking or running [2,3,18,21–23]. A comprehensive study in this field was conducted as a series of experiments by Shimazaki et al. [2,3,23,24].

Despite the relatively developed models to assess the mechanical behavior of footwear [25–27], there is currently no accepted model to justify the experimental results and predict the foot sole temperature. Recently, Nemati et al. [28] proposed an analytical method to predict the foot sole temperature during running and jogging at different speeds. In their model, they accounted for both the thermal and mechanical properties of the shoe sole. The model was then validated against experimental data and found to have a very good agreement between the measured foot sole temperatures and the predicted values. That model enables the designer to compare, for example, the softness of EVA12 and the superiority of the thermal properties of EVA08. The model has an analytical solution that can be used to accurately predict the arch area temperature for the initial duration of activity as up to 20 min during slow walking at 3 km/h and for up to 15 min for running at 9 km/h. However, after these times, the model over-predicts the foot sole temperature. Nemati et al. claimed that this may be due to the fact that the effect of sweating was not considered in their study [28]. Since the foot is capsulated inside the shoe, it is not possible to exchange heat to ambient environment by radiation or convection, efficiently. Therefore, sweating is the only effective temperature-controlling mechanism which relies on latent heat. While sweating is an effective factor in cooling feet, further care is required in modeling to assess the effect of sweating on the foot temperature.

The main aim of this study was to develop a model which can predict the temperature at the ball area and arch area in the first instance and to assess the accuracy of the model in predicting the foot sole temperature.

## 2. Model Development

The presented mathematical model is based on the following experimental scenario in which candidates wear the shoes without socks and stand still for a long time (around 10 min). At this time, the foot plantar surface and shoe sole reach thermal equilibrium, i.e., the shoe sole and foot plantar skin have the same temperature at their interface. Then, jogging commences for around 30 min. More detail of the experiment can be found in Section 3.

Based on the above, a transient conductive heat transfer equation can be considered for the shoe sole as follows [29]:

$$k_s \frac{d^2 T}{dx^2} + q_{Gen}''' = \rho_s c_{ps} \frac{dT}{d\tau} \tag{1}$$

where, $\rho_s$ and $c_{ps}$ are shoe sole density and specific heat properties, respectively. $q_{Gen}'''$ is the average heat generation rate per unit volume of the shoe sole due to viscous work during the stance phase of gait (Section 2.2). $T$ is the temperature at any point along the shoe sole thickness at any time, $\tau$.

Since the person has started the exercise from a standstill condition (please refer to Equation (12) in Section 2.3), therefore, there is a linear temperature distribution as the initial condition of Equation (1), i.e.,:

$$T(x,\ \tau = 0) = (T_\infty - T_{f0})\frac{x}{l} + T_{f0}, \dots 0 \le x \le l \tag{2}$$

The boundary conditions at the inner surface of the shoe sole and the outer surface of the shoe sole are, respectively:

$$\left[ q''_{met}(\tau) - q''_{sw} = -k_s \frac{dT}{dx} \right]_{x=0} \tag{3}$$

$$\left[ h(T - T_\infty) = -k_s \frac{dT}{dx} \right]_{x=l} \tag{4}$$

where, as discussed earlier, $q''_{met}$ and $h$ are the metabolism heat flux as a function of time and convective heat transfer coefficient, respectively, that will be discussed later. $q''_{sw}$ is the evaporative cooling rate due to sweating.

### 2.1. Convective Heat Transfer from Shoe Sole to Air

During walking/jogging, the outsole of the shoe is mainly cooled by air, forced convective heat transfer ($q''_{conv}$ in Figure 1) and the air average velocity over the shoe is equal to the person walking/jogging velocity (i.e. walking or jogging under zero wind condition). Considering, the outsole as a flat surface, the average convective heat transfer coefficient ($h$) can be calculated as [30]:

$$h = 0.037 \frac{k_a}{L} Re^{4/5} Pr^{1/3} \tag{5}$$

where $L$ is the shoe outsole length. In the above equation:

$$Re = \frac{\rho_a \cdot V \cdot L}{\mu} \tag{6}$$

$$Pr = \frac{\mu_a \cdot cp_a}{k_a} \tag{7}$$

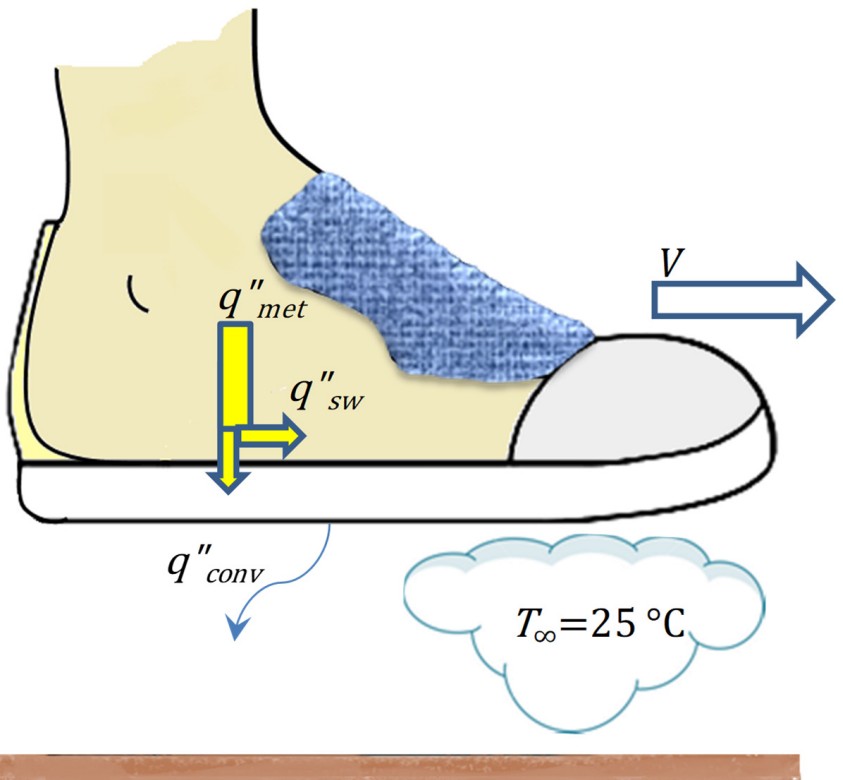

**Figure 1.** Different forms of heat transfer to/from shoe sole.

### 2.2. Heat Generation in Shoe Sole due to Periodic Loading–Unloading Condition

During the jogging, the shoe sole is periodically under the loading-unloading condition. The shoe sole is often constructed from viscoelastic materials (e.g., Polyurethane Elastomers, EVA foam, etc.) because of their shock-absorbent behavior [24,31]. For jogging activity, under the harmonic loading–unloading condition, viscous heat generates inside the shoe sole ($q'''_{Gen}$).

For this harmonic loading as shown in Figure 2, the volumetric generated heat during one period of the harmonic strain excitation is [32]:

$$U_d = \pi \varepsilon_0^2 E'' \tag{8}$$

$$\varepsilon_0 = \frac{\sigma_0}{\pi \sqrt{(E'')^2 + (E')^2}} \tag{9}$$

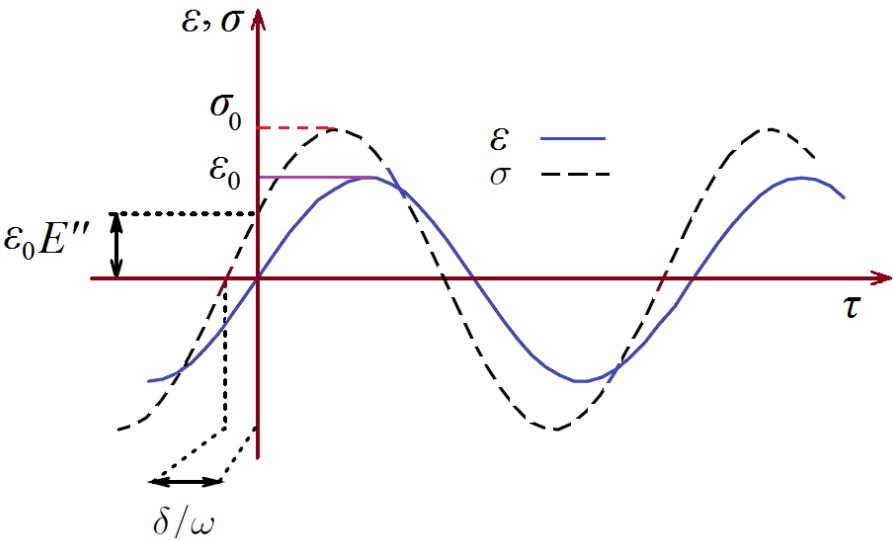

**Figure 2.** Harmonic strain excitation and stress response.

$E''$ and $E'$ are loss modulus and storage modulus, respectively. In addition, $\varepsilon_0$ and $\sigma_0$ and are strain and stress amplitudes, respectively.

Walking is also a harmonic motion. An example of the time-dependent gait motion pattern is presented in Figure 3, in which, the contact force (normalized by the person's weight) and the contact area for walking speed $V$ = 3 km/h can be considered. It is known that the maximum contact force value depends on the gait speed and it is higher at higher velocities. However, the average contact area is independent of gait speed [2]. Furthermore, the maximum contact force can be beyond the body's weight at foot landing during running [33].

The stress amplitude, $\sigma_0$, is equal to the maximum contact force divided by the average contact area. The average contact area is defined as the average foot sole area that is in contact with the shoe sole during walking.

Considering the above explanation, the volumetric generated heat can be calculated by dividing the volumetric generated heat (Equation (9)) by the period time ($\tau_{Stance} + \tau_{Swing}$):

$$q'''_{Gen} = \frac{U_d}{2(\tau_{Stance} + \tau_{Swing})} \cdot \frac{A_{cont}}{A} \tag{10}$$

For a harmonic loading, (Figure 2) a full period of a harmonic strain excitation includes both compression and expansion, while during walking, as shown in Figure 3, only the compressive loading is included (one-half of the harmonic cycle). Consequently, factor 2 appears in the denominator of Equation (10). Moreover, based on Figure 3, since the

contact area, $A_{cont}$, is different from the outsole area (the heat transfer area), Equation (10) is multiplied by $\frac{A_{cont}}{A}$.

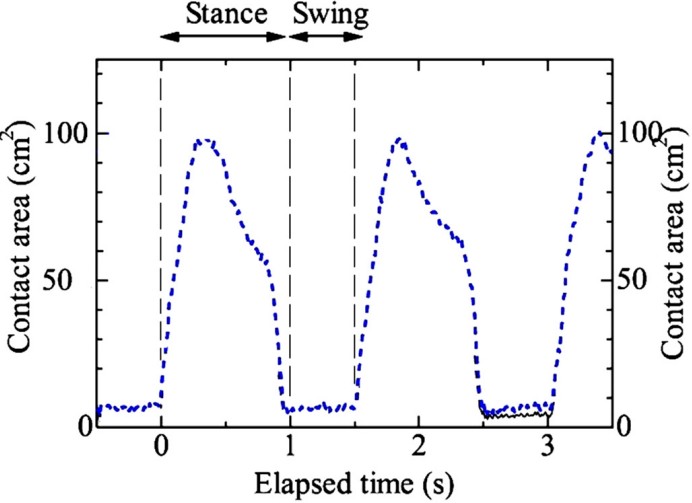

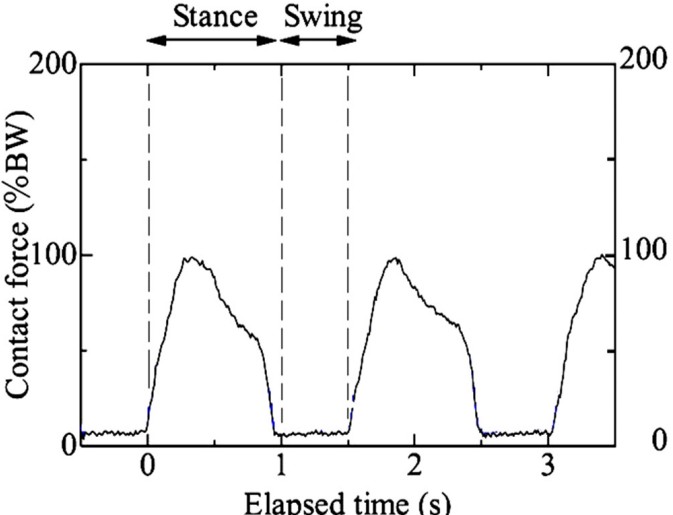

**Figure 3.** Shoe contact area and contact force for V = 3.0 km/h modified from [2].

### 2.3. Metabolism Rate Assessment during Walking/Jogging

The described experiment [3] was comprised of two steps. After wearing shoes and 10 min of rest, participants walked for 30 min. So, at the end of the first 10 min, it can be expected that the foot and shoe have reached a thermal equilibrium (Figure 4). In the absence of any mechanical work due to walking, and for this steady-state condition, the governing equation (Equation (1)) is reduced to Equation (11) with the following boundary condition.

$$k_s \frac{d^2 T}{dT^2} = 0$$
$$T(x = 0) = T_{f0}$$
$$T(x = l) = T_G$$
(11)

where $k_s$ is the shoe sole thermal conductivity and $T$ is temperature. $l$ is the shoe sole thickness. $T_{f0}$ and $T_G$ are the average foot sole temperature and ground temperature, respectively. Equation (11) is an ordinary differential equation with a solution in the form of:

$$T(x) = \left(T_\infty - T_{f0}\right)\frac{x}{l} + T_{f0}, \dots 0 \le x \le l$$
(12)

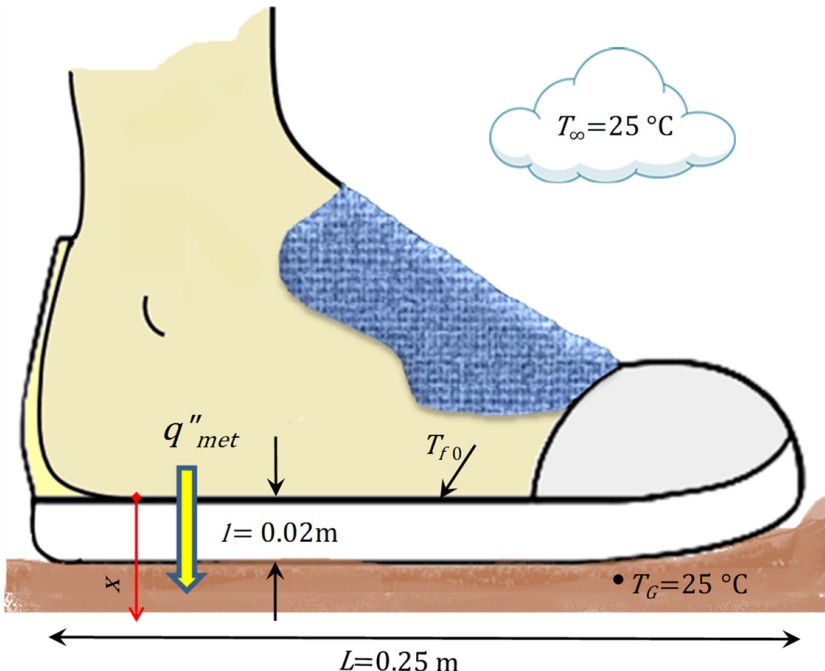

**Figure 4.** Thermal conditions for 10 min rest in the experiment.

This would be the initial condition at the begging of exercise.

Because of the thermal equilibrium and steady-state condition, the dissipated heat flux from the foot sole is equal to (Figure 4):

$$q''_{met0} = k_s \frac{T_{f0} - T_G}{l} \tag{13}$$

Starting from the rest condition to walking, the metabolism rate increases sharply by factor *F*:

$$q''_{met}(\tau) = F.q''_{met0} \tag{14}$$

The metabolism intensification factor (*F*) is a function of the gait speed and is adopted from Kipp's experimental correlation [34].

$$F(V) = 1.3 \frac{0.009V^2 + 0.002V + 0.082}{0.009V_{ref}^2 + 0.002V_{ref} + 0.082} \tag{15}$$

$V_{ref}$ is 0.833 m/s (3 km/h).

### 2.4. Latent Heat Loss due to Sweating during Walking/Jogging

The evaporation of sweat is an effective cooling mechanism for thermal regulation. Neither the sweating rate, nor the evaporation rate is uniform over the skin surface. It will be shown later that each part of foot sole has a different sweating evaporation. Cooling due to sweating is more considerable in the arc zone and is negligible in the ball of the foot zone. In the absence of more precise equations, the following equation is used to simulate the sweating–cooling effect:

$$q''_{sw} = h_{fg} h_e \times 4.7 \times 10^{-5} \times (P_{sat} - P)\left(T_f - T_{f0}\right) \tag{16}$$

Equation (16) was adopted from [35]. $h_{fg} = 2.43 \times 10^6$ J/kg is the water latent heat [36]. According to Equation (16), the sweating rate increases by increasing the temperature elevation $\left(T_f - T_{f0}\right)$. In the above equation, *P* is the vapour pressure in vicinity of the ball

zone (the area that sweating is considerable). $T_f$ is the foot sole skin temperature and $P_{sat}$ is the saturate vapour pressure at $T_f$. It was assumed that the vapour pressure at the skin is at the saturated pressure [37,38]. So, when the vapor pressure reaches the saturated pressure, evaporation stops. $P_{sat}$ can be estimated by the Antoine equation [36]:

$$P_{sat} = 100 \exp\left(18.956 - \frac{4030.18}{235 + T_f}\right) \tag{17}$$

However, the vapor pressure can be defined based on relative humidity, *RH*:

$$P = RH \cdot P_{sat} \tag{18}$$

and Equation (16) can be represented as:

$$q''_{sw} = h_{fg}h_e \times 4.7 \times 10^{-5} \times P_{sat}(1 - RH)\left(T_f - T_{f0}\right) \tag{19}$$

$h_e$ is the evaporative heat transfer coefficient, which is proportional to the square root of velocity [39]:

$$h_e = 0.000192 \times LR \times V^{0.5} \tag{20}$$

where *LR* is the Lewis ratio and approximately equals $16.5 \times 10^{-3}$ K/Pa.

## 3. Heat Transfer Modeling of a Shod Foot

The results of the current simulation were compared against the results from the literature [3]. The experiment was comprised of two stages [3]. In that experiment, first, participants wore the shoe with no socks, and after 10 min of rest, they did a jogging exercise for 30 min [3]. Foot skin temperatures were measured at various points every minute by thermocouples [3]. Participants were seven healthy adult males (1.72 ± 0.07 m height, 61.8 ± 4.7 kg weight, and 23.8 ± 4.3 years old) [3]. The footwear was a running shoe with a woven upper (Figure 5) [3]. The foot sole inside length is *L* = 25 cm (1 cm less than the shoe outsole length due to shoe thickness) [3]. The experiments were performed in a climate chamber with a constant air temperature of 25 °C, 58.5% *RH*, no solar radiation, and wind for three different speeds, i.e., 3, 6, and 9 Km/h [3]. More details can be followed in [3].

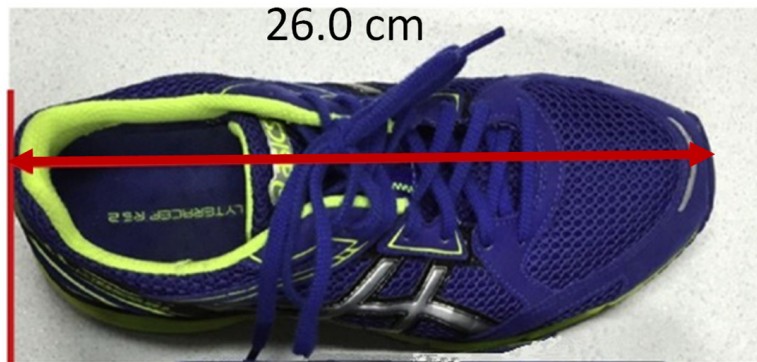

**Figure 5.** The running shoe used for the experiment (adapted from [3]).

The shoe sole was made-up of EVA08, and its properties are presented in Table 1:

**Table 1.** EVA08 thermal–mechanical properties based on what was reported in [23].

| $c_{ps}$ (J·kg$^{-1}$·K$^{-1}$) | $k_s$ (W·m$^{-1}$·K$^{-1}$) | $\rho_s$ (kg·m$^{-3}$) | $E''$ (Pa) | $E'$ (Pa) | Hardness |
|---|---|---|---|---|---|
| 2585 | 0.11 | 240 | $1.55 \times 10^6$ | $12.6 \times 10^6$ | A45 |

Thermo-physical properties of air are also presented in Table 2.

**Table 2.** Air thermo-physical properties at 25 °C based on what was reported in [30].

| Temperature (°C) | $c_{p_a}$ (J·kg$^{-1}$·K$^{-1}$) | $k_a$ (W·m$^{-1}$·K$^{-1}$) | $\rho_a$ (kg·m$^{-3}$) | $\mu_a$ (Pa·s) |
|---|---|---|---|---|
| 25 | 1006.96 | 0.0261 | 1.171 | $1.84 \times 10^{-5}$ |
| 35 * | 1007 | 0.0269 | 1.135 | — |

* is used in Section 4.

For this experiment, the average contact area is 0.006 m$^2$ for all gait speeds and the recorded maximum contact force is reported in Table 3 for each velocity. The average bare feet temperature at the beginning of the jogging was $T_{f0} = 34$ °C.

**Table 3.** Maximum contact force in percent of body weight (%B.W) for different gait speeds as reported in [2].

| Speed (km/h) | 3 | 6 | 9 |
|---|---|---|---|
| Max. Contact force %B.W. | 100 | 110 | 170 |

The equations were solved numerically using the Crank–Nicolson scheme. Since the set of equations is nonlinear, they had to be solved iteratively. To validate the numerical method, the problem described in [28] was solved numerically and compared with the proposed analytical solution in [28]. The maximum considered difference was less than 0.05%. The foot sole temperature rises $\left( T_f - T_{f0} \right)$ were calculated for three speeds with and without considering the sweating evaporative heat loss. The results are compared with experimental measurements based on [3], as presented in Figure 6.

A good agreement can be observed between the experimental and numerical results (Figure 6). According to Figure 6, the maximum increase in foot sole temperature after 30 min is 6 °C for a gait speed of 3 km/h. This value is 8 °C and 11.5 °C, for 6 km/h and 9 km/h, respectively.

Figure 6 shows that for a gait speed of 3 km/h, the effect of sweating on cooling is not strong and all curves, i.e., the numerical results with and without sweating and experimental measurements for ball and arch, are approximately overlapping. However, at higher speeds, especially for 9 km/h, the effect of sweating is very considerable. For a gait speed of 9 km/h, after the first 15 min, sweating as a thermoregulation mechanism comes into play and cools the arch area. The predicted temperature-time trend with sweating much closely replicate the temperature measured under the arch area while temperatures predicted without sweating more closely match the temperature-time trend at the ball of the foot.

For a gait speed of 6 km/h, sweating-induced cooling is triggered at a later time around 20 min, while at this speed the sweating at the ball zone seems to be negligible (Figure 6b).

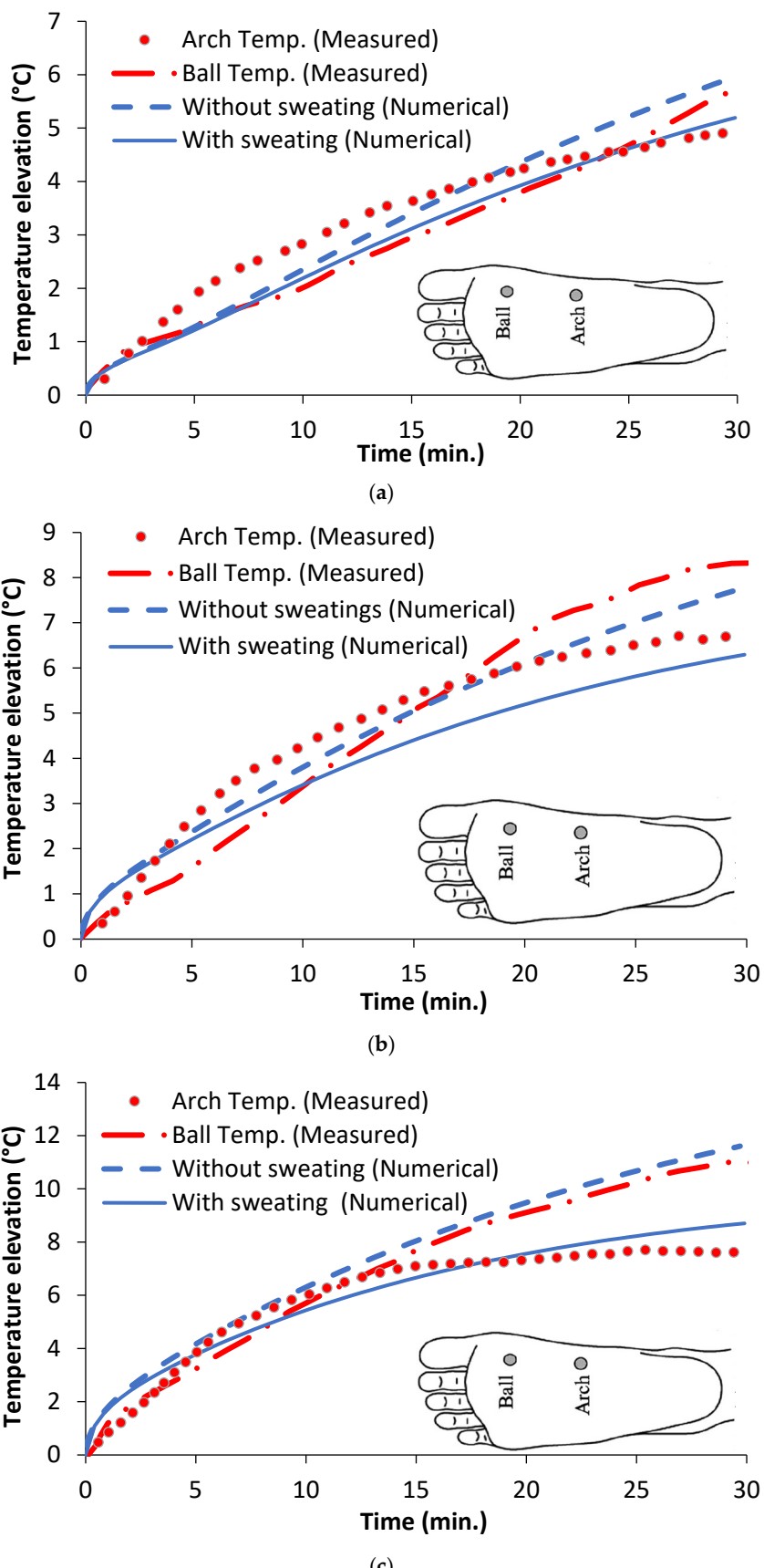

**Figure 6.** Comparison of foot sole temperature (numerical simulation and experimental measurements). (**a**) Gait speed 3 km/h; (**b**) Gait speed 6 km/h; and (**c**) Gait speed 9 km/h.

## 4. Illustrative Example

The results of this study show that sweating plays a principal role in the thermal regulation of the arch area. As an example of the application of the developed method, the effect of a porous turf-like insole design on foot temperature can be investigated. A schematic of the fibrous insole is shown in Figure 7. This insole is comprised of longitudinal fibers arranged in an aligned layout on a base. The thickness of the base is 1 mm and the length of the fiber is 10 mm. The fiber diameter is 1.2 mm (center to center distance) with the variable pitch.

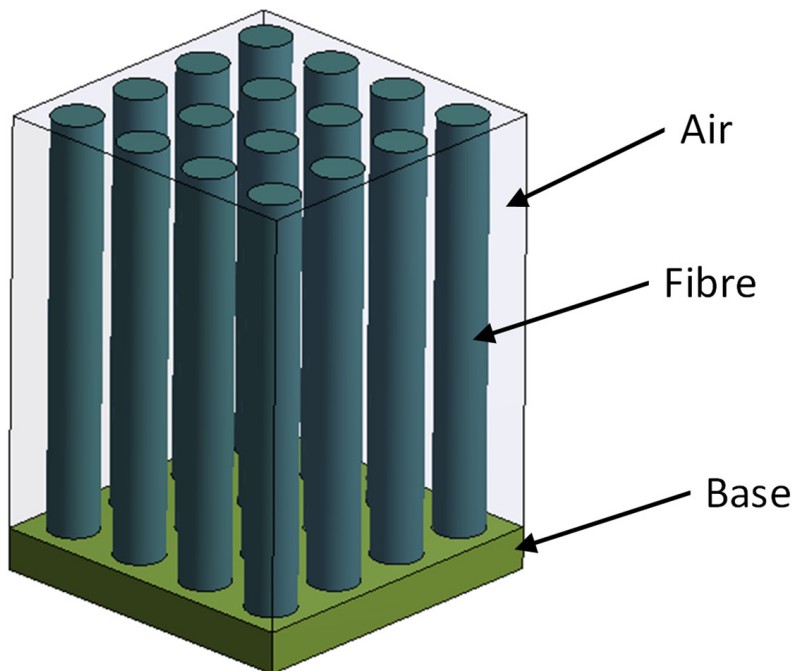

**Figure 7.** Schematic of the porous turf-like insole.

The insole is to be located inside the specified shoe for medical purposes. It was assumed that the insole material is thermoplastic polyurethane with the properties mentioned in Table 4.

**Table 4.** Thermoplastic polyurethane properties.

| $c_{pI}$ (J·kg$^{-1}$·K$^{-1}$) | $k_I$ (W·m$^{-1}$·K$^{-1}$) | $\rho_I$ (kg·m$^{-3}$) |
|---|---|---|
| 1700 | 0.258 | 1130 |

To consider the thermal properties of the insole, it was considered as a porous media, and properties were assumed as a volume-weighted average of air at 35 °C (Table 2) and thermoplastic polyurethane. To find the effective thermal conductivity, two different constant temperatures were assumed for the top and bottom of the insole and the 3D steady-state conduction heat transfer equation (Laplacian of temperature) was solved for two conjugated domains (air and thermoplastic polyurethane). Then, this was equated to a 1D heat transfer with the same temperature difference, the same geometry, and the same heat transfer rate to find the effective thermal conductivity. Results are presented in Table 5.

**Table 5.** Effective properties of the insole.

| Pitch (mm) | $c_{p_{eff}}$ (J·kg$^{-1}$·K$^{-1}$) | $k_{eff}$ (W·m$^{-1}$·K$^{-1}$) | $\rho_{eff}$ (kg·m$^{-3}$) |
|---|---|---|---|
| 1.5 | 1386.7 | 0.145 | 619.6 |
| 2 | 1248.1 | 0.096 | 393.9 |
| 2.5 | 1184.0 | 0.073 | 289.5 |
| 3 | 1149.2 | 0.060 | 232.7 |

Simulations were performed at speeds of 3 and 6 km/h. To find the extreme values of temperature, sweating was omitted. Results are presented in Figure 8.

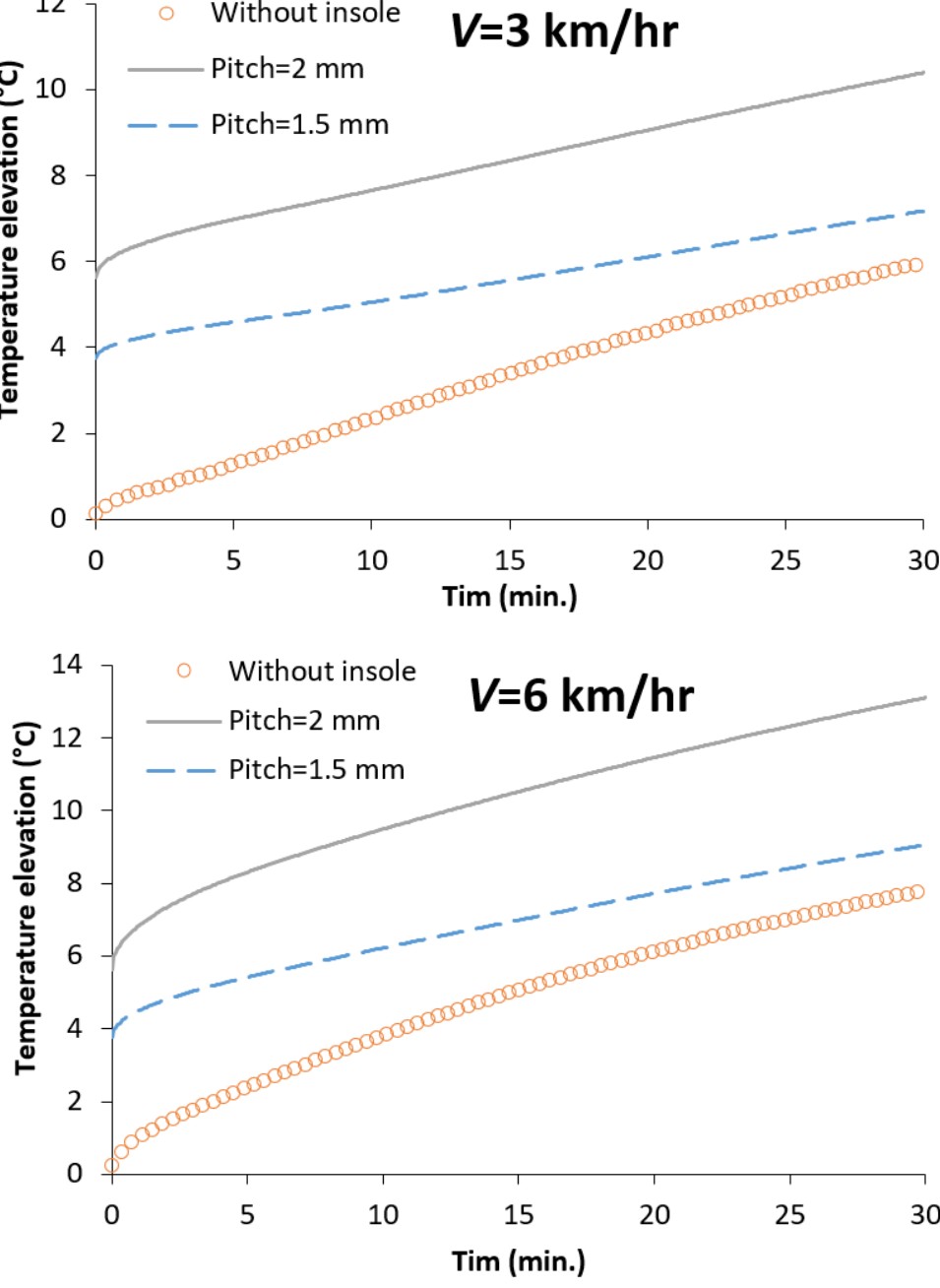

**Figure 8.** The foot sole temperature during shod gait at 3 and 6 km/h, using two porous turf-like insole designs with pitch of 2 mm and 1.5 mm and comparing with no insole condition.

Based on Figure 8, the presence of the insole causes a sharp temperature increase at the beginning of the activity, which is mainly due to trapped air between the fibers. This shows the importance of the presence of still air in keeping the foot warm. It was interesting to note that the difference between the "without insole" condition and the insole with a pitch of 1.5 mm decreases in the course of jogging. This shows the importance of insole heat capacity ($\rho_{eff}c_{p_{eff}}$), which prevents the temperature rise and keeps the foot sole temperature cool. So, in the design of an insole, both thermal conductivity and heat capacity must be considered.

## 5. Conclusions

An increase in the foot temperature raises the risk of plantar soft tissue injuries. So, the design of a comfortable shoe sole with proper thermal specifications is very important. However, the insole design is challenging as the effective parameters and the phenomena that regulated the foot temperature are complex. In this study, a comprehensive model was developed to predict the temperature of the barefoot sole in a shoe during walking and jogging. The results were validated against the experimental measurements. The effects of both the thermal and the mechanical properties of shoe soles were considered. Furthermore, the cooling effect due to sweat evaporation came into account as was presented at three different gait speeds. It was revealed that sweating is an effective parameter in foot arch zone thermal regulation while it is negligible at the ball of the foot area.

The presented method enables designers and researchers to investigate the interaction of different parameters and their impact before doing a large series of experiments. The study revealed the importance of a trapped layer of air in warming the foot sole and showed the importance of shoe sole thermal capacity besides its conductivity.

As an illustrative example, the effect of wearing a porous turf-like insole concept was investigated using the developed analytical model, where the foot sole temperature during the different levels of activities was predicted. This has shown that a different foot temperature can be achieved when the amount of material versus the air changes in the insole design. This developed model can potentially decrease the cost of experiments considerably by allowing an approximation of the insole design before producing the prototype.

As with any modelling approach, there are certain limitations that need to be taken into account. Since participants in the experiment were barefooted, the effect of sock properties such as material, thickness, and or humidity absorption can be considered in future studies.

**Author Contributions:** Conceptualization, H.N. and R.N.; methodology, H.N.; software, H.N.; writing—original draft preparation, H.N.; writing—review and editing, H.N and R.N. All authors have read and agreed to the published version of the manuscript.

**Funding:** This research received no external funding.

**Institutional Review Board Statement:** Not applicable.

**Informed Consent Statement:** Not applicable.

**Conflicts of Interest:** The authors declare no conflict of interest.

## Nomenclature

| | |
|---|---|
| $A$ | Area |
| $c_p$ | Specific heat |
| $E'$ | Storage modulus |
| $E''$ | Loss modulus |
| $F$ | Metabolism intensification factor |
| $h$ | Average convective heat transfer coefficient |
| $h_e$ | Evaporative heat transfer coefficient |
| $h_{fg}$ | Water latent heat |

| $k$ | Thermal conductivity |
|---|---|
| $L$ | Shoe length |
| $LR$ | Lewis ratio |
| $P$ | Pressure |
| $Pr$ | Prandtl number |
| $q''$ | Heat flux |
| $q'''$ | Volumetric heat generation rate |
| $Re$ | Reynolds number |
| $RH$ | Relative humidity |
| $T$ | Temperature |
| $l$ | Thickness |
| $U_d$ | Dissipated energy per unit of volume in one period |
| $V$ | Gait speed |
| $x$ | Cartesian coordinate axis |

Greek letters

| $\varepsilon_0$ | Strain amplitude |
|---|---|
| $\mu$ | Viscosity |
| $\rho$ | Density |
| $\sigma_0$ | Stress amplitude |
| $\tau$ | Time |

Subscripts

| $cont$ | Contact |
|---|---|
| $conv$ | Convection |
| $f$ | Foot sole in the jogging condition |
| $f0$ | Foot sole in the static steady condition |
| $G$ | Ground |
| $gen$ | Generation |
| $I$ | Insole |
| $met$ | Metabolism |
| $s$ | Shoe sole |
| $sat$ | Saturated |
| $sw$ | Sweating |
| $\infty$ | Ambient |

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
