# Peer review of "An Analytical Model to Predict Foot Sole Temperature: Implications to Insole Design for Physical Activity in Sport and Exercise"

_applsci, doi:10.3390/app12136806_

Round 1
Reviewer 1 Report
Authors presented the development a numerical and analytical model to estimate the foot sole temperature during activities. Though novelty and scientific sound are reported, there are several key issue should be properly addressed.
1. The measured temperature in the ball joint and arch regions were reported, while no information concerning the experimental protocol was found in the manuscript. Please add what instruments and how the placement was fixed for the temperature measurments?
2. The model validation part was missing in the manuscript, please add. This is a key part before this model could be of practical application.
3. Is there any limitation concerning this numerical model for temperature prediction?
Reviewer 2 Report
The research presented shows outstanding contributions to predict sole foot temperature. However, there are a few areas of improvement for this paper, which are listed below:
1. -The abstract does not enhance the novelty of the authors' work. Therefore, it is recommended to rewrite the main findings that denote the new research presented.
2. - What are the state-of-the-art findings regarding the prediction of sole foot temperature?
3. - These studies generally compare the accuracy of the experimental versus numerical analysis, but neither were developed evidence of both analyses. Could you mind discussing why data recollection was not analyzed in this research?
4.- What is the difference between the experimental model presented and the numerical analysis carried on?
5.- The authors have not discussed experimental results in detail. Therefore, there is no evidence that the experimental results have been collected for the solution obtained .
6.-It does not sufficiently describe how this research can improve the insole design for physical activity in sport and exercise.
7.- On the other hand, experimental and numerical analysis need to show evidence of similar results through graphics and images. Could you mind adding some graphics of the values reported for the thermal conductivity?
8.- My opinion is that the paper should show more explanation of the boundary conditions used during the experiments and simulations. The loads used and the gait cycles were used to measure the temperatures produced on the soles of the shoes.
9.- Why was not a method such as infrared thermography used to determine the thermal profile of the sole. What could other methods be used to assess temperature profiles in this research?
10.- Finally, the conclusion needs to recapitulate the significant findings of the experiment developed with future proposals to reduce or avoid temperature throughout the insole design
Reviewer 3 Report
Applied Sciences
Article
An analytical model to predict foot sole temperature: Implications to insole design for physical activity in sport and exercise
In this study, the authors have developed a model, it can predict the temperature at the ball area and arch area with reasonable accuracy. Since this mode has been considered most of the effective parameters and phenomena, it can be used as a standard for future experiments.
Well, articulated manuscript,
Adequate number of figures, tables,
Conclusions and Discussions are based on results
Grammar mistakes are minimal but can be improved.
Accept with minor changes.
Round 2
Reviewer 1 Report
Authors now address the concerns I had for the previous manuscript.
Reviewer 2 Report
In general, the improvements made by the authors in the revision look good. The reviewer appreciate the detailed response from the authors, which addressed the reviewer's major concerns, including the contribution of the work.